# Effects of Early versus Standard Central Line Removal on the Growth of Preterm Infants with Very Low Birth Weight: A Non-Inferiority, Randomized Clinical Trial

**DOI:** 10.3390/nu14224766

**Published:** 2022-11-11

**Authors:** Justyna Romańska, Tomasz Wawrzoniak, Paweł Krajewski, Joanna Seliga-Siwecka, Natalia Brunets, Izabela Lehman, Renata Bokiniec, Ewa Adamska, Barbara Królak-Olejnik, Jan Modzelewski, Tomasz Szczapa

**Affiliations:** 1Division of Neonatology and Neonatal Intensive Care, 1st Department of Obstetrics and Gynaecology, The Medical University of Warsaw, Starynkiewicza Sq. 1/3, 02-015 Warsaw, Poland; 2Department of Neonatology, Centre of Postgraduate Medical Education, Czerniakowska 231 St., 00-416 Warsaw, Poland; 3Department of Neonatology and Neonatal Intensive Care, The Medical University of Warsaw, Karowa 2 St., 00-315 Warsaw, Poland; 4Department of Prematurity and Neonatal Pathology, ŻELAZNA Medical Center Ltd. St. Sophia’s Specialist Hospital, Żelazna 90 St., 01-004 Warsaw, Poland; 5Department of Neonatology, Wroclaw Medical University, Borowska 213 St., 50-556 Wroclaw, Poland; 61st Clinic of Obstetrics and Gynecology, Centre of Postgraduate Medical Education, Żelazna 90 St., 01-004 Warsaw, Poland; 72nd Department of Neonatology, Neonatal Biophysical Monitoring and Cardiopulmonary Therapies Research Unit, Poznan University of Medical Sciences, Polna 33 St., 60-535 Poznan, Poland

**Keywords:** premature infant, central line, enteral nutrition, parenteral nutrition, growth, central-line-associated bloodstream infection

## Abstract

Very preterm infants are usually supported by parenteral nutrition delivered through central lines (CLs) while progressing with enteral intake, although the optimal time point for their removal is unclear. This study evaluated the impact of the CL discontinuation time on the short-term growth outcomes of preterm infants. A non-inferiority, parallel-group, randomized controlled trial was conducted in four neonatal intensive care units in Poland. Preterm infants with very low birth weight (VLBW) without congenital abnormalities were eligible. Patients were allocated to discontinue central access at an enteral feeding volume of 100 mL/kg/day (intervention group) or 140 mL/kg/day (control group). The study’s primary outcome was weight at 36 weeks’ postmenstrual age, with a non-inferiority margin of −210 g. Overall, 211 patients were allocated to the intervention or control groups between January 2019 and February 2021, of which 101 and 100 were eligible for intention-to-treat analysis, respectively. The mean weight was 2232 g and 2200 g at 36 weeks’ postmenstrual age in the intervention and control groups, respectively. The mean between-group difference was 32 g (95% confidence interval, −68 to 132; *p* = 0.531), which did not cross the specified margin of non-inferiority. No intervention-related adverse events were observed. Early CL removal was non-inferior to the standard type for short-term growth outcomes in VLBW infants.

## 1. Introduction

Optimizing nutrition is necessary to improve the neurodevelopmental and growth outcomes of preterm infants with very low birth weight (VLBW). The typical nutritional strategy for very preterm infants is that of gradually increasing the enteral feeding volume until full enteral intake is reached. During this period, infants are supported by parenteral nutrition delivered via central lines (CLs). By enabling administration of more concentrated parenteral nutrition for a prolonged period of time, central lines are still favored for optimal intravenous access in VLBW infants [1]. Studies have demonstrated early nutrition’s long-lasting effects on neurodevelopment, with lower nutrient intake correlated with poorer outcomes [2,3]. On the other hand, longer parenteral nutrition exposure increases the risk of infectious and mechanical complications associated with central venous catheters [4]. Central line-associated bloodstream infection (CLABSI) is one of the major complications of central catheters. Such infections are associated with increased mortality, short-term morbidity, longer duration of hospitalization, and higher hospital costs [4]. Observational studies have also shown a higher risk of long-term morbidity among neonatal sepsis survivors [5,6]. Care bundles have proved to be effective in reducing the rate of line-related sepsis in neonatal units [7]. Among other elements of care bundles, daily line-need assessment, complemented by removing central lines before the infant reaches full enteral intake, were components of quality improvement projects most effective in reducing CLABSIs [8]. However, the optimal time point for CL removal in relation to tolerated enteral milk volume remains uncertain because of the competing concerns mentioned above. Furthermore, given the limited evidence-based data, current clinical practice is mainly informed by expert opinions.

This pragmatic study aimed to evaluate the impact of the time when CL access was discontinued on the growth outcomes of preterm infants. We hypothesized that earlier CL removal, which is when an infant reaches 100 mL/kg/day of enteral intake (intervention group), would be non-inferior to its removal after reaching full enteral intake, defined as when an infant reaches 140 mL/kg/day of enteral intake (control group) regarding the weight of preterm infants at 36 weeks’ postmenstrual age (PMA). We selected a non-inferiority trial design, assuming that when earlier CL removal was non-inferior, it could tip the balance in its favor.

## 2. Materials and Methods

This study followed the reporting guidelines included in the Consolidated Standard of Reporting Trials (CONSORT) Statement and its extension for reporting non-inferiority and equivalence trials [9,10]. The study was prospectively registered on ClinicalTrials.gov (NCT03730883) and conducted according to a published protocol [11].

### 2.1. Trial Design

This was a multicenter, non-inferiority, parallel-group, randomized controlled trial conducted in tertiary neonatal intensive care units (NICUs) in Poland: Neonatal and Intensive Care Department, The Medical University of Warsaw; Division of Neonatology and Neonatal Intensive Care, The Medical University of Warsaw; Department of Reproductive Health, Centre of Postgraduate Medical Education, Warsaw; and Department of Neonatology, Wroclaw Medical University.

### 2.2. Participants

Eligible participants were all VLBW neonates with a central catheter whose birth weight was at or above the 3rd percentile at a given gestational age according to the 2013 Fenton Preterm Growth Chart [12]. There were no restrictions on the type of central catheter. Neonates with congenital illness or malformation that might affect growth or who may not survive were considered ineligible. After the parents signed the informed consent, infants receiving < 100 mL/kg/day of milk were randomly assigned to one of the two study groups.

### 2.3. Intervention

Patients in the intervention and control groups had their CLs removed after the infants reached 100 and 140 mL/kg/day of milk intake, respectively. The CLs were removed after three consecutive feedings of the targeted volume. Other aspects of parenteral and enteral nutrition followed the Polish Neonatal Society recommendations [13,14]. The decision to continue parenteral nutrition via the peripheral route was at the attending physician’s discretion. In addition, patients received routine clinical care in individual units.

### 2.4. Outcomes

This study’s primary outcome was weight at 36 weeks’ PMA. Secondary outcomes included length and head circumference at 36 weeks’ PMA, time to regain birth weight, all-cause death, CLABSI rate within the intervention period and 2 days after completion, the duration of hospital stay, the number of peripheral catheters inserted to continue parenteral nutrition after the intervention, and the need for CL insertion due to feeding intolerance within 7 days following intervention. CLABSI was defined according to the Centers for Disease Control and Prevention/National Healthcare Safety Network criteria [15]. In addition, the CLABSI rate was calculated by dividing the number of CLABSIs by the number of CL days and multiplying the result by 1000. We applied the criteria from the Vermont Oxford Network to classify the event as a bloodstream infection if only one blood culture was drawn and was positive for coagulase-negative *Staphylococcus* [16]. Other morbidities that occurred from birth to discharge home were compared between the groups, including the incidence of intraventricular hemorrhage grades 3 and 4 according to the criteria of Papile, periventricular leukomalacia (grade 2 or higher), bronchopulmonary dysplasia (oxygen therapy at 36 weeks’ PMA), Bell’s stage 2 or 3 necrotizing enterocolitis, patent ductus arteriosus (requiring medical treatment or surgical ligation), retinopathy of prematurity requiring treatment, and presumed (treated with antibiotics ≥ 5 days) or microbiologically confirmed late-onset sepsis. Apart from CLABSI, we monitored for the following CL-related adverse events (AEs): pleural, pericardial, and peritoneal effusions; cardiac tamponade; pericarditis; soft-tissue infiltration; and thrombosis. The list of AEs, which was supplemented with the open-ended section “other adverse events”, was an integral part of the electronic questionnaire. We recorded AEs occurring from birth until hospital discharge.

After the trial began, we added two more growth outcomes, which included growth velocity and change in Z-score for weight and head circumference, according to the Standardized Reporting of Neonatal Nutrition and Growth (StRONNG) checklist [17]. Specifically, the growth velocity was measured from birth to 36 weeks’ PMA using the exponential method [18]. The change in Z-score for weight and head circumference was calculated from birth to 36 weeks’ PMA based on the Fenton and Kim dataset using PediTools Web calculators [12,19].

The following instruments were used to perform anthropometric measurements: the electronic scale of the incubator, electronic baby scale, measuring board, and head circumference tape. The measurers followed the *Anthropometry Handbook* guidelines prepared by the INTERGROWTH-21st Anthropometry Group to obtain accurate, precise, and standardized readings [20]. Nutritional intake (enteral and parenteral) data for the first 4 weeks of life were collected. Mean weekly protein, lipid, carbohydrate, and energy intakes were calculated for weeks 1, 2, 3, and 4. Enteral feeds comprised the mother’s milk and donor human milk (with or without human milk fortifier) or preterm formula. We followed the recommendations presented in the StRONNG checklist to calculate the energy intake from the intravenous nutrition [17]. Calculations of enteral nutrients and energy intake from preterm formulas and human milk fortifiers were based on manufacturer information. We assumed that preterm breast milk provides 1.27 g, 3.46 g, 7.34 g, and 65.6 kcal per 100 mL of protein, fat, carbohydrate, and energy, respectively, and term breast milk and donor human milk provide 1.05 g, 3.9 g, 7.2 g, and 68 kcal per 100 mL of protein, fat, carbohydrate, and energy, respectively [21]. Enteral nutrients were considered 100% bioavailable, and calculations were based on the administered volumes to comply with the StRONNG checklist [17].

### 2.5. Sample Size

We calculated the mean ± standard deviation (SD) weight at 36 weeks’ PMA or the day of discharge as 2096 (357) g using retrospective data from 117 neonates with VLBW obtained in one of the study settings. The non-inferiority margin was set at −210 g, representing a 10% decrease in the mean body weight estimated based on retrospective data. We also considered the margin clinically justified because the value of 210 g is less than 1 SD below the mean for our historical cohort and is located within the one-centile range at 36 weeks’ PMA. Equal SD values for both groups were assumed for the sample size calculation. The sample size was estimated to detect a putative 50 g mean weight difference between both groups, as assessed by the non-inferiority test for two independent means (α = 0.025% and 80% power). Furthermore, we determined that 198 participants should be recruited by assuming equal-sized groups and a dropout rate of 20%.

### 2.6. Sequence Generation, Allocation Concealment, and Blinding

Participants were assigned to one of the two study groups based on a computer-generated randomization list. The randomization sequence was stratified by sex and gestational age (≤26 weeks + 6 days versus ≥ 27 weeks + 0 days) with a 1:1 allocation using blocks of varying sizes. A randomization sequence was created using an external statistical team. Allocation concealment was ensured as the statistical team released the assignment only after informed consent had been obtained from the eligible patient. Although the investigators and healthcare providers were aware of past assignments, the allocation schedule was concealed. Therefore, variable block sizes were chosen randomly from a specified subset of block lengths to prevent predictability and selection bias when using blocked randomization. Furthermore, neither the researchers nor the clinical team knew of the block lengths used.

The research and clinical teams were unblinded to group allocation because of the intervention’s nature. Similarly, the trial investigators who collected data and obtained anthropometric measurements, and also the data analysts, were aware of the trial-group assignments. Therefore, we used an objective and reliable primary outcome to minimize potential bias from the lack of blinding.

### 2.7. Statistical Analysis

The main hypothesis representing the primary outcome was tested for non-inferiority, whereas the other outcomes were tested for superiority. Statistical significance was assessed using a one-tailed, unpaired, two-sample Welch’s *t*-test, with a significance level set at 0.025 for the primary outcome. The non-inferiority margin was set at −210 g. We conducted intention-to-treat and per-protocol analyses of the primary outcome. Secondary growth and nutritional intake outcomes were analyzed using intention-to-treat, which were subject to data availability. In addition, all randomized patients were analyzed for major preterm complications, parenteral nutrition characteristics, and AEs, which were subject to data availability. AEs are presented using descriptive statistics. Dichotomous variables are summarized as numbers and percentages. Odds ratios (ORs) and the corresponding 95% confidence intervals (CIs) were calculated for dichotomous variables. Inter-group differences were tested using the χ^2^ test. Normally and non-normally distributed continuous variables are presented as means ± SDs and medians (interquartile ranges), respectively. The mean difference and 95% CI are presented for normally distributed continuous outcomes. Means between the treatment groups were compared for significance using Welch’s *t*-test. For non-normally distributed continuous outcomes, the median difference and 95% CI are presented. Differences between the groups were compared using the Mann–Whitney test. Significant differences were considered at *p* < 0.05, or when the CI excluded 0 for the mean/median difference or 1 for the OR. All analyses were performed using the SciPy Python package.

## 3. Results

### 3.1. Trial Participants

Overall, 211 neonates from four study centers were recruited between January 2019 and February 2021. Five infants were withdrawn due to necrotizing enterocolitis, spontaneous intestinal perforation, or other abdominal surgery, but their data were analyzed based on initial assignments. Parental consent for participation and data collection was withdrawn from one patient. In addition, five and four infants in the early- and standard-removal groups died before 36 weeks’ PMA, respectively. The remaining patients were included in the intention-to-treat analysis of the primary outcomes (Figure 1). In the early-removal group, 16 patients in the intention-to-treat analysis were excluded from the per-protocol analysis, including 13 who did not receive the allocated intervention because of clinician preference, 1 who discontinued the intervention due to an AE, and 2 who were withdrawn before intervention. Conversely, 16 patients did not receive the allocated intervention because of clinician preference, and 4 discontinued the intervention because of AEs in the standard-removal group. Therefore, 85 and 80 infants in the intervention and control groups, respectively, were suitable for the per-protocol analysis. No clinically relevant differences between the two trial groups were observed in the baseline characteristics (Table 1).

### 3.2. Primary Outcomes

The mean weight at 36 weeks’ PMA in the intervention and control groups was 2232 g and 2200 g, respectively. The mean difference in weight at 36 weeks’ PMA between the early- and standard-removal groups was 32 g (95% CI, −68 to 132; *p* = 0.531), which did not exceed the specified non-inferiority margin of −210 g (Figure 2). The per-protocol analysis yielded similar results: the mean weight at 36 weeks’ PMA in the intervention and control groups was 2225 g and 2180 g, respectively (mean difference, 76 g; 95 CI, −38 to 189; *p* = 0.19).

### 3.3. Secondary Outcomes

No significant differences between the two study groups were observed regarding the time to regain birth weight, growth velocity, weight, length, or head circumference at 36 weeks’ PMA. The early- and standard-removal groups showed decreased Z-scores for weight and head circumference between birth and 36 weeks’ PMA. In addition, the early-removal group experienced a mean change in Z-score for head circumference of −0.71, compared to −0.32 in the standard-removal group (mean difference, −0.39; 95% CI, −0.69 to −0.09; *p* = 0.011). Changes in Z-score for weight, weight and head circumference < 10th percentile at 36 weeks’ PMA were similar between the two groups. Table 2 summarizes the secondary growth outcomes.

Furthermore, no significant between-group differences were observed in mortality or major preterm complications (Table 3).

Table 4 presents the weekly mean protein, lipid, carbohydrate, and energy intakes for the first month of life. Notably, there were slight but significant differences in the mean protein, carbohydrate, and energy intake in week 1.

For infants in the intervention group, the median number of central line days was 7.5 days, compared with 8 days in the control group (median difference, −0.5; *p* = 0.064), and the duration of parenteral nutrition delivered through peripheral access after the intervention was 2 days and 1 day, respectively (median difference, −1; *p* = 0.163) (Table 5).

### 3.4. Adverse Events

Table 6 shows the AEs that occurred in the study population. CL-associated bloodstream infections were the most frequent AEs in both groups. The CLABSI rate was 10.35 and 10.2 in the early- and standard-removal groups, respectively. One infant in the intervention group presented with sudden cardiac instability from a CL-related cardiac tamponade on the first day of life. However, the infant was successfully resuscitated with favorable outcomes at hospital discharge. In the control group, one infant developed peritoneal effusion and two others experienced soft-tissue infiltration. Catheter-tip malposition was found in the above cases. No difference was found between the study groups concerning the number of CLs reinserted during the 7 days after the intervention. Infants in the early-removal group were twice as likely to have a peripheral line inserted after the intervention to continue parenteral nutrition (56% and 27.5%, respectively). We found no difference in the median number of peripheral lines inserted per patient between the study groups.

## 4. Discussion

In this pragmatic, multicenter, randomized trial involving infants with VLBW, CL removal at 100 mL/kg/day of milk proved to be non-inferior to that at 140 mL/kg/day, concerning the weight of infants at 36 weeks’ PMA. Earlier CL removal also did not affect in-hospital growth velocity or most growth outcomes, which were assessed at 36 weeks’ PMA, excluding the greater fall in Z-score for head circumference in the intervention group. This observation is of particular concern because of the consistent positive association between postnatal head growth and neurodevelopmental outcomes reported in observational studies [22]. The current strategy to improve very preterm infants’ growth is to initiate parenteral nutrition and amino acid supplementation on the first postnatal day [23]. The macronutrient intake analysis for the first 4 weeks of life in our study cohort showed a minimal but significant decrease in the mean protein and energy intakes during week 1. Nevertheless, both groups achieved recommended protein and energy intakes [23,24]. A meta-analysis, including the early versus late introduction of parenteral nutrition studies, found that early parenteral nutrition effectively improved short-term growth outcomes, except for the head circumference [25]. In addition, a meta-analysis comparing low- and high-dose and early and late parenteral amino acid administration found no differences in short-term growth or long-term neurodevelopmental outcomes [26]. Contrary to these findings, a recent large cohort study’s results by Roze et al. found a significant positive association between high early amino acid intake (˃3.5 g/kg/day) and cognitive outcomes at the age of 5 years [27]. Therefore, considering these conflicting data, a conclusion cannot be drawn regarding the causal effect of the control group’s slightly increased early nutrient intake and improved in-hospital head growth. Moreover, a measurement bias may have occurred since head circumference measurements were obtained at birth by the nursing staff, who attended deliveries and were performed during the infant’s initial stabilization. Delivery room management of preterm infants using equipment for thermal control and respiratory support may conflict with obtaining precise measurements.

The only study to date that investigated the timing of CL removal regarding in-hospital growth outcomes was that by Perrem et al. [28]. In this study, infants with VLBW were randomly assigned to CL removal and parenteral nutrition discontinuation at two different volumes of enteral intake: 100 mL/kg/day versus 140 mL/kg/day of milk, with time to regain birth weight as the primary outcome. The authors found that early CL removal caused a slower regain of birth weight. The mean difference of 1.5 days was statistically significant but smaller than what the authors predefined as clinically significant. Indeed, it may be argued whether a difference of such magnitude is clinically important since the threshold for this clinical outcome has not been established. Here, the between-group mean difference of 0.5 days in the time to regain birth weight was not statistically significant. However, our findings, which demonstrate similar growth parameters at 36 weeks’ PMA between the two study groups, are consistent with those of the study reported by Perrem et al.

Although earlier CL removal confers potential advantages in reducing CLABSIs, we did not demonstrate a lower incidence of CLABSIs in the intervention group. We suspect that the reasons for this could be two-fold. First, it is not a single intervention, but multiple practices implemented simultaneously as care bundles, which are proven to effectively prevent CLABSIs [7]. Second, no obvious differences were observed between the study groups concerning days with CL access.

Since twice as many patients in the intervention group continued parenteral nutrition through peripheral access compared with those in the control group, these differences should be carefully considered. It could be stated that the advantage of decreasing the duration of central access might be balanced by the risk associated with peripheral access. Notably, the decision to continue parenteral nutrition through the peripheral line was left to the neonatal caregivers’ discretion. Therefore, it cannot be excluded that it reflects a specific nutritional strategy in the participating units. An approach to providing parenteral nutrition either through central or peripheral lines until an infant attains full enteral intake has recently been challenged by the concept of exclusive enteral feeds, which is safe and feasible in stable infants with VLBW [29].

This trial had several strengths that should be noted. First, using appropriate methodology, the study answers a general question that neonatologists encounter in their everyday practice. Second, this multicenter trial design enhanced the results’ generalizability. Third, we followed the recommendations from the StRONNG checklist to improve the quality of reporting nutrition and growth outcomes in our study [17]. Finally, since a simple intervention was applied to an almost unselected population of infants with VLBW, we believe it can be easily implemented in NICUs.

This study had some limitations. First, according to the study protocol, infants with a congenital anomaly that might have affected growth were considered ineligible, whereas those with gastrointestinal conditions requiring surgery were withdrawn. Therefore, our findings cannot be generalized to more selected population. Second, the study was unblinded because of the nature of the intervention. Thus, the possibility that knowledge of trial-group assignments could impact clinician practice cannot be excluded. Third, the primary outcome assessors were unblinded to the trial group assignments. We consider this unlikely to affect the objective primary outcome used in the study. Fourth, the non-inferiority margin may be debatable since it was chosen arbitrarily after analyzing epidemiological data from one of the study centers. Finally, weight at 36 weeks’ PMA is only a short-term growth outcome and not predictive of later cognitive impairment, as recently reported by Fenton et al. [30]. However, our findings provide a background for further studies to examine early CL removal and later neurocognitive outcomes.

## 5. Conclusions

Among VLBW infants, earlier CL removal did not negatively impact in-hospital growth. Consequently, the risk of maintaining the CL until the infant reaches full enteral intake is not balanced by improved growth at 36 weeks’ PMA. Therefore, this study’s results may provide nutritional guidelines and affect clinical practice.

## Figures and Tables

**Figure 1 nutrients-14-04766-f001:**
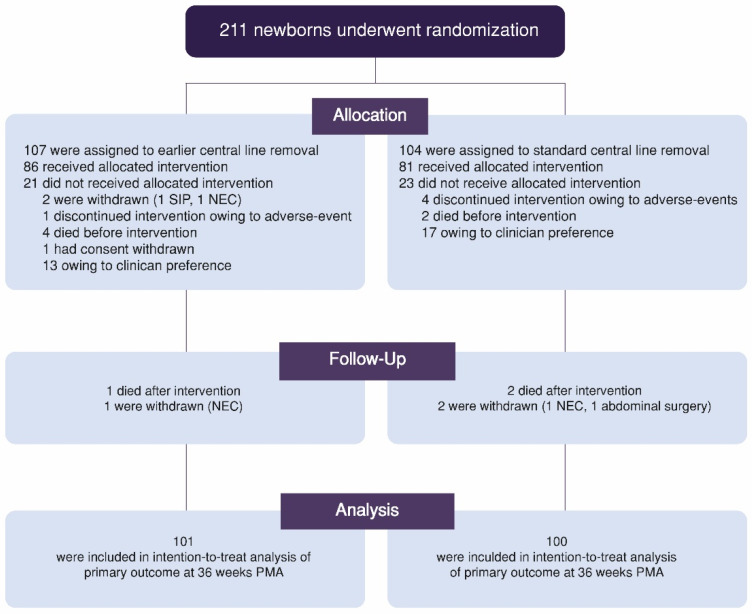
Consolidated Standard of Reporting Trial flow diagram. Abbreviations: PMA, postmenstrual age; SIP, spontaneous intestinal perforation; NEC, necrotizing enterocolitis.

**Figure 2 nutrients-14-04766-f002:**
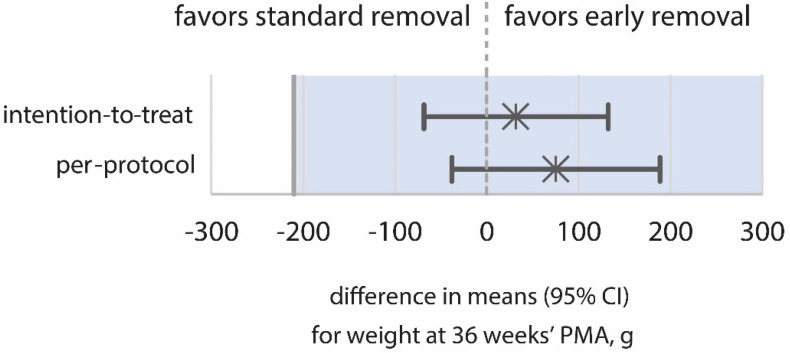
The mean difference in weight at 36 weeks’ PMA between the early- and standard-removal groups. A solid line at the mean difference of −210 g indicates a non-inferiority margin; the area to the right of the mean difference of −210 g indicates values for which early central line removal is non-inferior to the standard central line removal. Abbreviations: PMA, postmenstrual age; CI, confidence interval.

**Table 1 nutrients-14-04766-t001:** Patients’ baseline characteristics.

Characteristic	Early-Removal Group, n = 106	Standard-Removal Group, n = 104
Female sex—n (%)	49 (48.5%)	49 (49.0%)
Gestational age at delivery—mean ± SD, wk	28.3 (2.3)	28.7 (2.3)
Birth weight—mean ± SD, g	1122 (259)	1139 (233)
Birth weight < 1000 g—n (%)	30 (29.7%)	30 (30.0%)
Birth weight < 10th percentile of gestational age—n (%)	5 (5.0%)	7 (7.0%)
Z-score for birth weight—mean ± SD	−0.12 (0.82)	−0.29 (0.82)
Apgar score at 5 min—median (IQR)	8.0 (2.0)	7.0 (1.0)
Cesarean delivery—n (%)	82 (81.2%)	91 (91.0%)
Received antenatal steroids, any—n (%)	99 (98.0%)	95 (95.0%)
Multiple pregnancy, any—n (%)	37 (36.6%)	47 (47.0%)
	n = 100	n = 99
HC at birth—mean ± SD, cm	26.3 (2.5)	26.4 (2.2)
HC at birth < 10th percentile of gestational age—n (%)	7 (6.9%)	11 (11.0%)

Abbreviations: HC, head circumference; IQR, interquartile range; SD, standard deviation.

**Table 2 nutrients-14-04766-t002:** Primary and secondary growth outcomes.

Outcome	Early-Removal Group n = 101	Standard-Removal Group n = 100	Effect Measure (95% CI) *	*p*-Value
Primary outcome				
Weight at 36 weeks’ PMA—mean ± SD, g	2232 (364)	2200 (356)	32 (−68 to 132)	0.531
Secondary outcomes				
	n = 101	n = 100		
Z-score for weight at 36 weeks’ PMA—mean ± SD	−1.05 (0.86)	−1.13 (0.87)	0.08 (−0.16 to 0.32)	0.517
Change in Z-score for weight from birth to 36 weeks’ PMA—mean ± SD	−0.94 (0.62)	−0.84 (0.64)	−0.09 (−0.27 to 0.08)	0.302
Weight < 10th percentile at 36 weeks’ PMA—n (%)	44 (43.6%)	42 (42.0%)	1.07 (0.61 to 1.86)	0.935
Time to regain birth weight—mean ± SD, days	10.62 (5.21)	10.14 (4.92)	0.48 (−0.93 to 1.89)	0.499
Growth velocity from birth to 36 weeks’ PMA— mean ± SD, g/kg/day	13.6 (2.3)	13.8 (2.5)	−0.2 (−0.8 to 0.5)	0.645
	n = 89	n = 94		
HC at 36 weeks’ PMA—mean ± SD, cm	31.7 (1.6)	31.8 (1.3)	−0.1 (−0.5 to 0.3)	0.567
Z-score for HC at 36 weeks’ PMA—mean ± SD	−0.54 (1.01)	−0.44 (0.89)	−0.10 (−0.38 to 0.18)	0.479
Change in Z-score for HC from birth to 36 weeks’ PMA—mean ± SD	−0.71 (1.13)	−0.32 (0.93)	−0.39 (−0.69 to −0.09)	0.011
HC < 10th percentile at 36 weeks’ PMA—n (%)	16 (18.0%)	19 (20.2%)	0.85 (0.41 to 1.79)	0.816
	n = 91	n = 95		
Length at 36 weeks’ PMA—mean ± SD, cm	43.7 (2.2)	44.0 (2.1)	−0.3 (−0.9 to 0.3)	0.313
Z-score for length at 36 weeks’ PMA—mean ± SD	−1.27 (0.83)	−1.11 (0.81)	−0.15 (−0.39 to 0.09)	0.208
Length < 10th percentile at 36 weeks’ PMA—n (%)	44 (48.4%)	39 (41.1%)	1.34 (0.75 to 2.40)	0.393

Abbreviations: HC, head circumference; PMA, postmenstrual age; CI, confidence interval; SD, standard deviation. * Odds ratios and mean differences are shown for binary and continuous outcomes, respectively.

**Table 3 nutrients-14-04766-t003:** Mortality and major preterm complications.

Event	Early-Removal Group, n = 106	Standard-Removal Group, n = 104	Odds Ratio (95% CI)	*p*-Value
	n (%)			
Death before discharge	5 (4.7%)	4 (3.8%)	1.24 (0.32–4.74)	1.000
Necrotizing enterocolitis, Bell’s stage 2 or 3	6 (5.7%)	4 (3.8%)	1.50 (0.41–5.48)	0.769
Spontaneous intestinal perforation	3 (2.8%)	0 (0.0%)	6.03 (0.30–121.86)	0.252
Other abdominal surgeries	1 (0.9%)	2 (1.9%)	0.49 (0.04–5.44)	0.987
RDS treated with a surfactant	70 (66.0%)	67 (64.4%)	1.07 (0.61–1.90)	0.920
BPD—oxygen dependency at 36 weeks’ PMA	15 (14.2%)	12 (11.5%)	1.26 (0.56–2.85)	0.719
PDA requiring medical treatment or surgical ligation	21 (19.8%)	28 (26.9%)	0.67 (0.35–1.28)	0.291
Retinopathy of prematurity requiring treatment	13 (12.3%)	5 (4.8%)	2.77 (0.95–8.07)	0.092
Metabolic bone disease	10 (9.4%)	14 (13.5%)	0.67 (0.28–1.58)	0.484
Intraventricular hemorrhage grade 3 or 4	9 (8.5%)	6 (5.8%)	1.52 (0.52–4.42)	0.619
Cystic periventricular leukomalacia	5 (4.7%)	5 (4.8%)	0.98 (0.28–3.49)	1.000
Early-onset sepsis	3 (2.8%)	1 (1.0%)	3.00 (0.31–29.32)	0.627
Late-onset sepsis other than CLABSI	39 (36.8%)	29 (27.9%)	1.51 (0.84–2.70)	0.218

Abbreviations: RDS, respiratory distress syndrome; BPD, bronchopulmonary dysplasia; PDA, patent ductus arteriosus; CLABSI, central line-associated bloodstream infection; PMA, postmenstrual age; CI, confidence interval.

**Table 4 nutrients-14-04766-t004:** Nutritional intake in weeks 1–4.

Nutritional Intake, Mean ± SD	Early-Removal Group, n = 95	Standard-Removal Group, n = 96	Mean Difference (95% CI)	*p*-Value
Protein intake—g/kg/d				
Week 1	2.99 (0.31)	3.11 (0.23)	−0.12 (−0.20 to −0.04)	0.002
Week 2	3.41 (0.46)	3.44 (0.44)	−0.03 (−0.16 to 0.10)	0.607
Week 3	3.56 (0.63)	3.66 (0.57)	−0.10 (−0.28 to 0.07)	0.234
Week 4	3.56 (0.56)	3.71 (0.48)	−0.16 (−0.31 to −0.01)	0.041
Lipid intake—g/kg/d				
Week 1	2.69 (0.58)	2.79 (0.63)	−0.09 (−0.27 to 0.08)	0.291
Week 2	4.86 (1.15)	5.06 (1.08)	−0.19 (−0.51 to 0.12)	0.232
Week 3	5.61 (0.95)	5.83 (0.87)	−0.23 (−0.49 to 0.03)	0.089
Week 4	5.71 (0.87)	5.91 (0.71)	−0.19 (−0.42 to 0.03)	0.095
Carbohydrate intake—g/kg/d				
Week 1	10.92 (1.17)	11.32 (1.18)	−0.40 (−0.74 to −0.07)	0.019
Week 2	14.47 (1.77)	14.98 (1.46)	−0.51 (−0.98 to −0.05)	0.031
Week 3	14.93 (1.70)	15.10 (1.48)	−0.17 (−0.63 to 0.28)	0.454
Week 4	14.92 (1.43)	14.86 (1.48)	0.06 (−0.36 to 0.47)	0.790
Energy intake—kcal/kg/d				
Week 1	76.3 (8.5)	79.1 (8.8)	−2.8 (−5.3 to −0.3)	0.028
Week 2	112.9 (15.9)	116.3 (13.8)	−3.4 (−7.6 to 0.9)	0.121
Week 3	123.2 (14.8)	126.3 (13.4)	−3.2 (−7.2 to 0.9)	0.124
Week 4	124.4 (13.6)	126.8 (10.2)	−2.4 (−5.8 to 1.0)	0.171

Abbreviations: CI, confidence interval; SD, standard deviation.

**Table 5 nutrients-14-04766-t005:** Characteristics of parenteral nutrition.

Characteristic	Early-Removal Group, n = 100	Standard-Removal Group, n = 102	Effect Measure (95% CI) *	*p*-Value
Days with a central catheter—median (IQR)	7.5 (5.0)	8.0 (5.0)	−0.5	0.064
Patients with central catheters reinserted during 7 days after intervention—n (%)	3 (3.0%)	3 (2.9%)	1.02 (0.20–5.18)	1.000
Patients with peripheral catheters inserted after intervention—n (%)	56 (56.0%)	28 (27.5%)	3.36 (1.87–6.05)	0.000071
Number of peripheral catheters per patient inserted after intervention—median (IQR)	2.0 (1.3)	2.0 (1.0)	0.0	0.517
Duration of PN delivered through the peripheral catheter after intervention—median (IQR)	2.0 (1.3)	1.0 (1.0)	1.0	0.163

Abbreviations: PN, parenteral nutrition; IQR, interquartile range; CI, confidence interval. * Odds ratios and median differences are shown for binary and continuous outcomes, respectively.

**Table 6 nutrients-14-04766-t006:** Adverse events.

Event	Early-Removal Group, n = 106	Standard-Removal Group, n = 104
CLABSI-rate-events/1000 catheter-days	10.35	10.20
Cardiac tamponade—n (%)	1 (0.9%)	0
Peritoneal effusion—n (%)	0	1 (1.0%)
Soft-tissue infiltration—n (%)	0	2 (1.9%)

Abbreviations: CLABSI, central line-associated bloodstream infection.

## Data Availability

The data presented in this study are available upon request from the corresponding author.

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
