# Peer review of "Effects of Early versus Standard Central Line Removal on the Growth of Preterm Infants with Very Low Birth Weight: A Non-Inferiority, Randomized Clinical Trial"

_nutrients, 2022, doi:10.3390/nu14224766_

Round 1

Reviewer 1 Report

Thank you for the opportunity to revise the manuscript entitled: "Effects of Early versus Standard Central Line Removal on the Growth of Preterm Infants with Very Low Birth Weight: A Non-Inferiority, Randomized Clinical Trial ".

 This study evaluated the impact of the CL discontinuation time on the short-23 term growth outcomes of preterm infants.

 The manuscript is very interesting and it could be accepted before only to review some minor English language revision and amplify the Introduction section.

The quality of the tables is very satisfactory. The reference list exhaustively covers the relevant literature in an unbiased way. Statistical methodologies are valid and coherent with the aim of the study.

All the methodology section is sufficiently documented in order to allow replication studies.

The manuscript follows a high rigor in its structure, high quality in the writing and in the quality of the content.

I believe that the manuscript will capture the interest of the audience interesting in this field.

Reviewer 2 Report

The aim of this multicentre study is to evaluate the impact of the time of discontinuation of central parenteral nutrition on short-term growth outcomes of preterm infants. The data in the paper show that early removal of central parenteral nutrition did not fall short of the guidelines for short-term growth outcomes in preterm infants.

The study was conducted in a formally correct manner. There are some minor issues to be addressed before possible publication. 

Table 1. shows Patients' baseline characteristics. Why is a statistical difference between the two groups not shown?

Introduction:

what are the advantages of early parenteral removal? reduction of infections, costs etc.?

Discussion:

are there studies that suggest the use of concomitant enteral and parenteral in these patients?

What is the role of maternal breastfeeding in these early patients?

Are there differences between caesarean and natural childbirth also with regard to the patients' microbiota?
